# Antiproliferative and Antimigration Activities of Beauvericin Isolated from *Isaria* sp. on Pancreatic Cancer Cells

**DOI:** 10.3390/molecules25194586

**Published:** 2020-10-08

**Authors:** Hiroaki Yahagi, Tadahiro Yahagi, Megumi Furukawa, Keiichi Matsuzaki

**Affiliations:** School of Pharmacy, Nihon University, 7-7-1 Narashinodai, Funabashi, Chiba 274-8555, Japan; phhi17001@g.nihon-u.ac.jp (H.Y.); yahagi.tadahiro@nihon-u.ac.jp (T.Y.); manyu0907@hotmail.com (M.F.)

**Keywords:** beauvericin, EMT, entomopathogenic fungi, *Isaria*, isariotin, migration, pancreatic cancer

## Abstract

This study describes the antiproliferative and antimigration effects of beauvericin from a culture broth of *Isaria* sp. in human pancreatic cancer cells (PANC-1). Activity-guided fractionation of the EtOAc extract of cultured broth of *Isaria* sp. RD055140 afforded beauvericin (**1**), a new isariotin derivative, 7-*O*-methylisariotin C (**2**), together with the known isariotin analogs, TK-57-164A (**3**) and B (**4**). As a result of the measurement of the cell viability, **1** inhibited cell growth (IC_50_ = 4.8 µM) of PANC-1 cells. Furthermore, **1** was found to inhibit the migration activity of PANC-1 cells by upregulating the expression of the *E-cadherin* gene and reducing *N-cadherin* and *Snail* genes in a dose-dependent manner (0.1–1 µM). These activities of **1** had lower concentrations than the cytotoxic activity. These findings suggest that **1** can be used as an anticancer agent against human pancreatic carcinoma.

## 1. Introduction

Pancreatic cancer is one of the most common tumors worldwide, but its malignancy leads to poor prognosis. Basal treatment is surgical resection, but it is difficult to find in early stages due to few subjective symptoms and the ease of distant metastasis. As a result, it can be managed with this treatment only in limited cases [1,2]. Because the majority of cases are found in advanced stages, this cancer responds poorly to currently available medical therapies. Hence, it is necessary to develop novel anticancer agents to inhibit cancer proliferation or metastasis.

Some reports suggest that epithelial to mesenchymal transition (EMT) plays an important role in tumor progression and metastasis, including pancreatic cancer. EMT occurs by loss of epithelial cells and acquisition of mesenchymal characteristics by stimulation of some factors, such as transforming growth factor (TGF-β), fibroblast growth factor (FGF) and hypoxia-inducible factor (HIF), and so on. During EMT progression, the expression of epithelial markers, e.g., E-cadherin, occludin or cytokeratin, is downregulated, while mesenchymal markers, such as N-cadherin, vimentin and fibronectin are upregulated. Moreover, transcriptional factors, such as Snail, Slug, Zeb-1, and Twist1, are associated with the regulation of the expression of these markers, so it is important to control such factors to suppress the EMT process of pancreatic cancer [3,4].

Entomopathogenic fungi are parasitic to some species of insects. These fungi kill the host insect to acquire nutrients and grow into a fruiting body. They are used as a portion of healthy food and traditional medicine in Asian countries. They produce various secondary metabolites with a wide variety of biological activities, such as anticancer [5], immunosuppressant and cardiovascular [6].

We hypothesize that compounds that have an inhibitory activity on cell growth and EMT could be candidates for anticancer drugs against pancreatic cancers, which have cells that can easily acquire abilities of migration and invasion. Based on a cytotoxicity screening of pancreatic cancer cell lines, the EtOAc extract of *Isaria* sp. RD055140 showed strong inhibitory activity. In this report, for the isolation of active compounds from *Isaria* sp. and the evaluation of the activity (cell viability: 31.4%, 10 μg/mL, positive control: cisplatin, 69.2%, 100 µM), we describe the structural analysis of four compounds and the assessment of their active mechanisms toward EMT.

## 2. Results

### 2.1. Structures of **1**–**4**

The EtOAc extract of *Isaria* sp. RD055140 exhibiting inhibitory activity of cell growth was fractionated on bioassay-guided fractionation to afford **1**–**4** (Figure 1).

Compound **1** was identified as beauvericin by comparison of its NMR and MS spectra with those reported [7].

Compound **2** was obtained as a colorless amorphous solid and had [α]_25D_ −46.5 (c 0.10, CH_3_OH). The molecular formula of **1** was determined to be C_22_H_33_NO_7_ by HR-ESI-MS (*m*/*z* 446.2155 [M + Na]^+^, calcd. for 446.2155), corresponding to 8 degrees of unsaturation. IR absorption bands suggested an amino functional group at 3390 cm^−1^, hydroxyl functional groups at 3298 cm^−1^, amide and keto carbonyls at 1748, 1708, 1674, 1635 cm^−1^ and an ether functional group at 1082 cm^−1^. The ^1^H-NMR spectrum displayed one methyl proton signal at δ_H_ 0.90 (t, *J* = 7.0 Hz), seven methylenes at δ_H_ 1.27 (m), δ_H_ 1.32 (m), δ_H_ 1.55 (m), δ_H_ 1.71 (m), δ_H_ 2.19 (q, *J* = 7.0 Hz), δ_H_ 2.44 (t, *J* = 7.0 Hz), δ_H_ 2.49 (t, *J* = 7.5), one methine at *δ*_H_ 3.05 (ddd, *J* = 1.0, 3.5, 7.5), two olefins at *δ*_H_ 6.01 (d, *J* = 15.5 Hz) and another at *δ*_H_ 6.78 (m), one methoxy at δ_H_ 3.48 (s), four oxymethines at δ_H_ 3.49 (dd, *J* = 1.5, 3.5 Hz), δ_H_ 3.48 (s), δ_H_ 3.88 (d, *J* = 7.5 Hz) and δ_H_ 4.30 (q, *J* = 2.0 Hz) and one *N*-methine at δ_H_ 4.73 (dq, *J* = 2.5, 13.0 Hz). The ^13^C-NMR, DEPT, and HMQC spectra showed 22 carbon signals, which could be assigned one methyl signal at δ_C_ 14.28, eight methylenes at δ_C_ 23.37, δ_C_ 23.52, δ_C_ 24.57, δ_C_ 32.26, δ_C_ 32.52, δ_C_ 38.59, δ_C_ 42.49, δ_C_ 43.52, one methine at δ_C_ 57.25, two olefinic methines at δ_C_ 125.06 and another at δ_C_ 145.39, four oxygenated methines at δ_C_ 65.16, δ_C_ 56.46, δ_C_ 72.04, δ_C_ 78.71, one *N*-methine at δ_C_ 47.67, one oxygenated quaternary carbon at δ_C_ 77.25, two keto carbonyls at δ_C_ 208.77 and another at δ_C_ 213.56, one amide at δ_C_ 167.78 and one methoxy at δ_C_ 58.64. The HMBC spectrum revealed that proton signals at δ_H_ 2.09 (t, *J* = 13.0 Hz, H-3b) and δ_H_ 2.24 (ddd, *J* = 1.5, 5.0, 13.0 Hz, H-3a) both correlated to the methylene at δ_C_ 38.59 (C-2), which indicated these proton signals were non-equivalent. The ^1^H–^1^H COSY spectrum clarified the hydrogen sequence in bold, as shown in Figure 2. The interrelationships of these partial structures were determined by HMBC experiment. The HMBC correlations from proton signals at δ_H_ 4.30 (H-1) to carbon signals δ_C_ 208.77 (C-9), from δ_H_ 2.24 and δ_H_ 2.09 (H-3) to δ_C_ 77.25 (C-4) and C-9, from δ_H_ 3.49 (H-5) to C-9, from δ_H_ 3.88 (H-7) to δ_C_ 56.46 (C-6) and C-5 and from δ_H_ 3.05 (H-8) to C-6 and C-9 indicated that this molecule had the bicyclo[3.3.1]nonane carbon skeletal structure (C-1–C-9). The ^1^H-NMR spectrum also revealed mutually coupled signals at H-5 and H-6, which correlated to the methine carbon signals at C-5 and C-6, respectively, in the HMQC spectrum. The chemical shifts of these signals showed the presence of an epoxide functional group in this molecule. Other HMBC cross-peaks from δ_H_ 6.01 (H-2′) to δ_C_ 167.78 (C-1′), from δ_H_ 2.49 (H-6′) to δ_C_ 213.56 (C-7′) and from δ_H_ 2.44 (H-8′) to δ_C_ 213.56 (C-7′) indicated this molecule had a side chain composed of 12 carbons. The side chain was located at C-2 by HMBC correlation from H-2 to amide carbonyl at C-1′. The HMBC correlation between H-7 and the methoxy carbon at δ_C_ 58.64 indicated the location of a methoxy group at C-7. As a result of these spectral analyses, the structure of **2** was proposed as 7-*O*-methylisariotin C, which is a new compound.

Compounds **3** (TK-57-164A) and **4** (TK-57-164B) were found to be constitutional isomers of **2** when the NMR and MS spectra were compared with those previously reported [8,9].

### 2.2. Measurement of Cell Viability

To evaluate the potential effect of **1***–***4** on human pancreatic cancer cellular proliferation, we used the human pancreatic cancer cell line PANC-1. An MTT proliferation assay demonstrated that **2**–**4** did not show an inhibitory effect on cell growth, but **1** decreased the viability of PANC-1 at the concentration of 1–10 µM (IC_50_ = 4.8 μM) in a dose-dependent manner (Figure 3).

### 2.3. Measurement of Cell Migration by Wound-Healing Assay

We performed a wound-healing assay to evaluate whether **1** can inhibit not only cell viability, but also the cell migration of PANC-1. After 48 h of incubation with 10 ng/mL TGF-β and 0.1*–*1 µM of **1**, the motility of PANC-1 cells was inhibited, whereas the cells treated with 10 ng/mL TGF-β and vehicle DMSO migrated through the wounded area to close the wound (Figure 3). Furthermore, we investigated the effect of **1** on the expression of EMT marker genes in PANC-1 cells in the presence of TGF-β. In Figure 4, cells treated with 10 ng/mL TGF-β for 48 h. led to an increase in the mRNA of EMT markers such as *N-cadherin*, *Snail*, and *Slug* and a decrease in *E-cadherin*, so TGF-β induced EMT by signal transduction in PANC-1 cells. Furthermore, cells treated with TGF-β plus 1 μM of **1** upregulated the expression of *E-cadherin* and decreased that of *N-cadherin* and *Snail*.

## 3. Discussion

The present study elucidates the effects of four compounds isolated from *Isaria* sp. RD055140 for cytotoxicity and migration and the mechanisms of migration activity on cells of the pancreatic cancer cell line PANC-1. Beauvericin (**1**), but not isariotin analogs (**2**–**4**), showed significant antiproliferative activity. In addition, **1** inhibited cell migration in a dose-dependent manner by the mechanisms of upregulation of *E-cadherin* and downregulation of *N-cadherin* and *Snail* genes.

EMT occurs frequently in the progression of various malignancies like migration and invasion [10]. As TGF-β is released from the marginal cells of tumor cells, it is associated with this process and enhances the activity of migration [11]. The regulation of the TGF-β signal pathway is effective in inhibiting cancer progression. We evaluated the effect of **1** treatment on the migration ability using a wound-healing assay to find that 0.1–1 µM of **1** significantly inhibited migration of PANC-1 cells with a lower dose of cytotoxic activity (IC_50_ = 4.8 µM). Our results (Figure 3) showed **1** had the ability of antagonistic activity toward TGF-β signaling. However, the underlying mechanisms at this point are unclear, and further studies are needed to clarify this.

It is known that TGF-β signaling downregulates E-cadherin, occludin and cytokeratin, while upregulates N-cadherin, vimentin, fibronectin, Snail, Slug, Zeb-1 and Twist1 to progress EMT [11]. This results in a loss of epithelial phenotypes and the acquisition of mesenchymal phenotypes and leads to EMT. Consequently, regulating these factors is an important key to inhibiting tumor progression, so we evaluated the effect of **1** for EMT-related factors by RT-PCR. Our results showed that **1** inhibited the downregulation of *E-cadherin* and the upregulation of *N-cadherin* and *Snail* genes, while little influenced *Zeb-1*, *Vimentin* and *Tbr1*. These results indicate that **1** could regulate not only epithelial but also mesenchymal genes.

Beauvericin is a cyclic hexadepsipeptide that belongs to the enniatin family produced by various fungi and has many biologic activities, especially anticancer effects [12]. Beauvericin has been reported to induce the apoptosis of human non-small cell lung cancer cells and human epidermoid carcinoma cells through mitochondrial pathways, including the decrease of reactive oxygen species, loss of mitochondrial membrane potential, the release of cytochrome c and activation of caspase-9 and -3 [13,14]. Many reports have been published about its anticancer effects, including apoptosis and oxidative stress in vitro. However, studies focusing on the inhibition of cell migration have not been reported [7], and this mechanism has not been examined. Based on our research, this study is the first report that describes not only cytotoxicity but also the antimigration activity and the mechanisms of the inhibitory effect of beauvericin on EMT via the TGF-β-signaling pathway on PANC-1 pancreatic cancer cells. Both beauvericin and enniatins have antibiotic and ionophoric properties and different bioactivities [15]. Reports that enniatin affects cell migration have not been published. Therefore, it is necessary to further investigate the structure-activity relationship between beauvericin and similar compounds and the further precise mechanisms underlying the inhibitory effect on EMT.

Isariotins were originally isolated from the cultured broth of *I. tenuipes,* and many derivatives have been isolated from *Isaria* species [16]. Because some isariotin analogs have been reported to exhibit cytotoxic activities against human epidermoid carcinoma cells, small cell lung cancer cells and human breast cancer cells [17,18], they are considered as the candidates for novel anticancer agents. We isolated three isariotin analogs, including one new compound in this report, but unfortunately, they have no effects against the viability of PANC-1 cells. Thus, we hope to investigate isariotin derivatives, which have inhibitory activities on cell growth and cell migration on PANC-1 cells.

In conclusion, this study has demonstrated for the first time that **1** inhibits TGF-β-induced EMT by regulating EMT-related genes in pancreatic cancer. Because of the difficulty of early detection, pancreatic cancer is often considered one of the poorest-prognosis cancers compared to any other cancer type. Many patients need to take an anticancer agent, for example, gemcitabine, 5-FU, cisplatin or paclitaxel; however, these drugs are not effective enough [11]. Resolving this problem is a key factor in improving the prognosis of pancreatic cancer. As such, our research provides a possible lead to the development of a new anticancer agent that has both inhibitory activities of cell growth and EMT.

## 4. Materials and Methods

### 4.1. General Procedures

UV and IR spectra were recorded using a V-730 BIO spectrophotometer and an FT/IR-4200 spectrophotometer (JASCO, Tokyo, Japan), respectively. Optical rotations were determined using a P1020 polarimeter (JASCO, Tokyo, Japan). NMR spectra were measured using a JNM ECX-500 NMR and JNM ECX-600 NMR spectrometer (JEOL, Tokyo, Japan). ESI mass spectra were obtained using a Xevo G2-S QTOF (Waters, Milford, MA, USA).

Dimethyl sulfoxide (DMSO), 10× phosphate-buffered saline (PBS) and agar were purchased from FUJIFILM Wako Pure Chemical Corporation (Osaka, Japan). 3-[4,5-Dimethylthiazol-2-yl]-2,5-diphenyl-tetrazolium (MTT) was purchased from Dojindo Molecular Technologies, Inc. (Kumamoto, Japan). Methanol-*d*_4_ was purchased from Kanto Chemical Co., Inc. (Tokyo, Japan). Cisplatin was purchased from Sigma-Aldrich Co. LLC (St. Louis, MO, USA). TGF-β was purchased from BioLegend, Inc. (San Diego, CA, USA). Roswell Park Memorial Institute media (RPMI 1640) was purchased from Nacalai Tesque (Kyoto, Japan). Fetal bovine serum and penicillin-streptomycin were purchased from Gibco (Grand Island, NY, USA). Difco^TM^ potato dextrose broth (PDB) and Difco^TM^ potato dextrose agar (PDA) were purchased from Becton, Dickinson and Company (Franklin Lakes, NJ, USA).

### 4.2. Cell Culture

PANC-1, a human pancreatic cancer cell line, was purchased from RIKEN BioResource Research Center (Ibaraki, Japan) and was cultured in RPMI 1640 supplemented with 10% FBS and 1% penicillin-streptomycin at 37 °C in an atmosphere of 5% CO_2_.

### 4.3. Measurement of Cell Viability by MTT Assay

PANC-1 cells were plated on a 96-well plate at 1 × 10^5^ cells/mL and incubated. After 24 h, the medium was changed and treated with various concentrations of the samples. Control cells were treated with DMSO. After incubation of the cells with samples for 48 h, 20 μL of 1 mg/mL MTT solution was added to each well and incubated at 37 °C for four hours. The supernatant was then removed and 150 μL DMSO was added to each well to dissolve the formazan. The formazan formation was measured with a microplate reader at a wavelength of 570 nm and a reference wavelength of 655 nm.

### 4.4. Extraction and Isolation of Chemical Compounds from Isaria sp. RD055140

*Isaria* sp. RD055140 that was purchased from the National Institute of Technology and Evaluation (NITE) Biological Resource Center (NBRC) was precultured in a PDA medium for seven days in a 10 cm Petri dish. Cultured agar medium was hollowed out by a cork borer (7 mm i.d.), inoculated in 300 mL or 500 mL baffled Erlenmeyer flasks containing 100 mL or 150 mL of PDB medium and cultured for seven days at 25 °C on a rotary shaker at 150 rpm.

The cultured broth (10.5 L) was extracted with the same volume of EtOAc and filtered. The EtOAc layer was concentrated in vacuo to give the EtOAc extract (1.14 g). The extract was chromatographed on preparative silica gel TLCs developed with CHCl_3_-CH_3_OH-H_2_O (90:12:1), and each fraction was extracted with CH_3_OH (250 mL) to obtain 10 fractions (fr. 1 to 10). The third fraction (202.5 mg) was purified by HPLC (column: YMC Pack Pro C18 (10 mm i.d. × 250 mm), flow-rate: 3.0 mL/min) using a gradient program of CH_3_OH-H_2_O (7:3)/0 min→(9:1)/60 min as an eluate to give **1** (22.2 mg). The fourth fraction was purified on HPLC using the same column, eluted with a gradient of CH_3_OH-H_2_O (6:4)/0 min→(9:1)/60 min and yield **2** (3.5 mg), **3** (5.6 mg) and **4** (5.6 mg).

*Beauvericin* (**1**): colorless amorphous solid, [α]_25D_ +6.14 (c 0.1, CH_3_OH); HR-ESI-MS *m*/*z* [M + Na]^+^ 806.3987 (calcd. for C_45_H_57_N_3_O_9_Na, 806.3990); ^1^H- and ^13^C-NMR (600 MHz, 150 MHz, CD_3_OD): See Appendix A.

*7*-*O*-*methylisariotin C* (**2**): colorless amorphous solid, [α]_25D_ −46.5 (*c* 0.1, CH_3_OH); HR-ESI-MS *m*/*z* [M + Na]^+^ 446.2155 (calcd. for C_22_H_33_NO_7_Na, 446.2155); UV λ_max_ (CH_3_OH, logε) 208 nm (4.25), FT-IR (KBr) 3390, 3289, 2939, 1748, 1708, 1674, 1635, 1082 cm^–1^; ^1^H- and ^13^C-NMR (500 MHz, 125 MHz, CD_3_OD): See Appendix A.

*TK*-*57*-*164A* (**3**): colorless amorphous solid, [α]_25D_ −7.57 (*c* 0.1, CH_3_OH); HR-ESI-MS *m*/*z* [M + Na]^+^ 432.2361 (calcd. for C_22_H_35_NO_6_Na, 432.2362); ^1^H- and ^13^C-NMR (500 MHz, 125 MHz, CD_3_OD): See Appendix A.

*TK*-*57*-*164B* (**4**): colorless amorphous solid, [α]_25D_ −6.90 (*c* 0.1, CH_3_OH); HR-ESI-MS *m*/*z* [M + Na]^+^ 404.2053 (calcd. for C_20_H_31_NO_6_Na, 404.2049); ^1^H- and ^13^C-NMR (500 MHz, 125 MHz, CD_3_OD): See Appendix A.

#### 4.4.1. Measurement of Cell Migration by Wound-Healing Assay

PANC-1 cells were seeded in 24-well plates at 2 × 10^5^ cells/mL and cultured for 48 h until they reached confluence. Then, a linear wound was generated in the monolayer with a sterile 200 μL micropipette tip. Any cellular debris was removed twice with PBS and incubated with serum-free RPMI 1640 medium with samples together with 10 ng/mL TGF-β. Control cells were treated with DMSO and with or without TGF-β. After incubation for 24 and 48 h, the cells were photographed with a digital camera, and the migrated area was measured using NIH Image J (ver. 1.51) software.

#### 4.4.2. Reverse Transcription-Polymerase Chain Reaction (RT-PCR)

The expression levels of *E-cadherin*, *N-cadherin*, *Snail*, *Slug*, *Zeb-1*, *Vimentin* and TGF-β receptor 1 (*Tbr1*) mRNA were determined using RT-PCR. After incubation on a 24-well plate in a sample-containing medium for 48 h, total RNAs were extracted using an RNeasy Mini Kit (Qiagen, Hilden, Germany), following the manufacturer’s instructions. Reverse transcription reactions were performed with a PrimeScript RT-PCR Kit (Takara Bio, Inc., Shiga, Japan) following the manufacturer’s instructions, using 250 ng of total RNA. The expression levels of the above genes were confirmed using *Gapdh* as a control.

The primers had the following sequences: for *E-cadherin*, forward: 5′-GCCTCCTGAAAAGAGAGTGGAAG-3′, reverse: 5′-TGGCAGTGTCTCTCCAAATCCG-3′, for *N-cadherin*, forward: 5′-GAGGAGTCAGTGAAGGAGTCA-3′, reverse: 5′-GGCAAGTTGATTGGAGGGATG-3′, for *Snail*, forward: 5′-TGCCCTCAAGATGCACATCCGA-3′, reverse: 5′-GGGACAGGAGAAGGGCTTCTC-3′, for *Slug*, forward: 5′-ATCTGCGGCAAGGCGTTTTCCA-3′, reverse: 5′-GAGCCCTCAGATTTGACCTGTC-3′, for *Slug*, forward: 5′-ATCTGCGGCAAGGCGTTTTCCA-3′, reverse: 5′-GAGCCCTCAGATTTGACCTGTC-3′, for *Zeb-1*, forward: 5′-AAGTAACCCTGTGTATTTCTGGATGA-3′, reverse: 5′-TGGGATCAACCACCAATGG-3′, for *Vimentin*, forward: 5′-AGGCAAAGCAGGAGTCCACTGA-3′, reverse: 5′-ATCTGGCGTTCCAGGGACTCAT-3′, for *Tbr1*, forward: 5′-CGCACTGTCATTCACCAT-3′, reverse: 5′-AAACCTGAGCCAGAACCT-3′ and for *Gapdh*, forward: 5′-ACGGATTTGGTCGTATTGGG-3′, reverse: 5′-TGATTTTGGAGGGATCTCGC-3′.

### 4.5. Statistical Analysis

The data are expressed as the means ± standard error (SE). Quantification was done using NIH Image J ver. 1.51 (National Institutes of Health, Rockville, Maryland, USA). Statistical comparisons among different treatment groups were performed using one-way analysis of variance (ANOVA) with Dunnett’s multiple comparison test using JMP 14 (SAS Institute Inc., Cary, NC, USA). The level of statistical significance was set at *p* < 0.05 or 0.01.

## Figures and Tables

**Figure 1 molecules-25-04586-f001:**
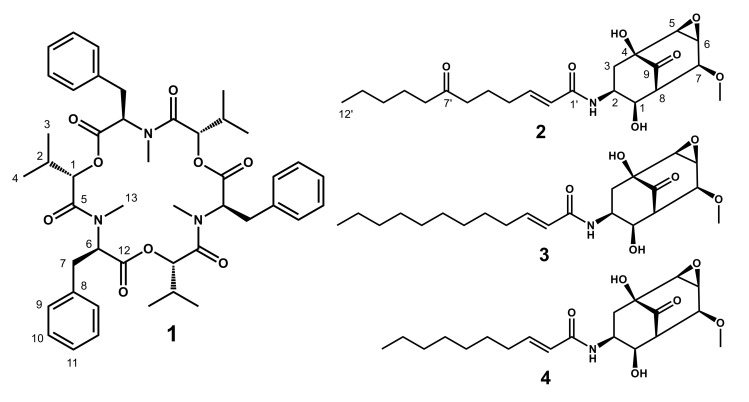
Structures of isolated compounds from the EtOAc extract of the culture medium of *Isaria* sp. RD055140.

**Figure 2 molecules-25-04586-f002:**
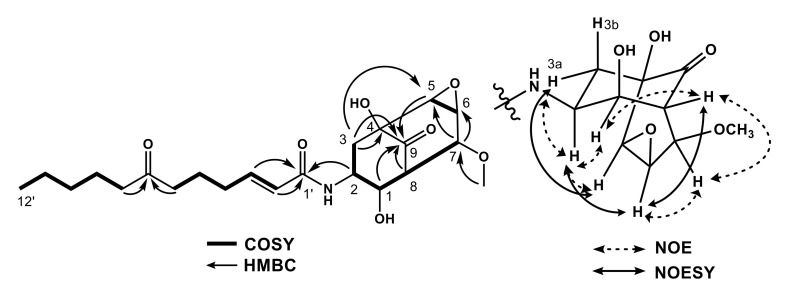
Key correlations of 2D NMR and relative configuration for **2**.

**Figure 3 molecules-25-04586-f003:**
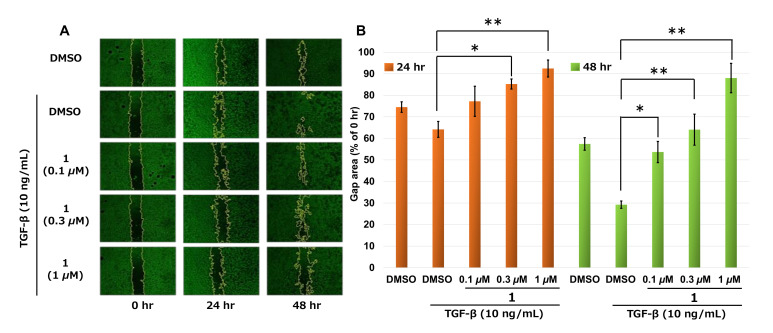
Effect of **1** on the migration of PANC-1 cells. (**A**) Wound-healing assay, PANC-1 cells grown at confluency. The monolayers were scratched with a 200 µL pipette tip in the central area of the culture. TGF-β and **1** were added at the indicated times. Photographs were taken using phase-contrast microscopy (magnification at 40×). (**B**) Quantification of the cell-free area. Quantification was done using NIH Image J (ver. 1.51). Statistical analysis was performed using JMP 14. Data are expressed as the mean ± SE. These experiments were done in triplicate (N = 3). * *p*< 0.05, ** *p*< 0.01.

**Figure 4 molecules-25-04586-f004:**
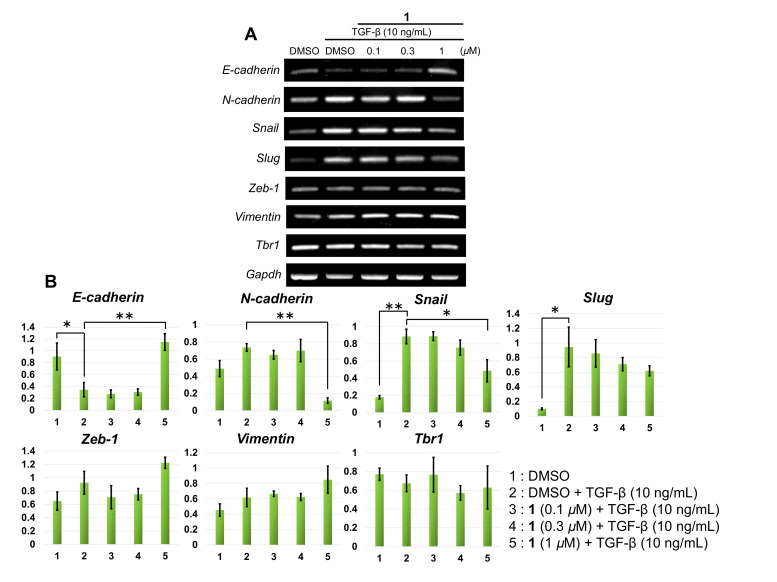
Compound **1** inhibited epithelial to mesenchymal transition (EMT) by regulating the expression of EMT markers. (**A**) The expression of mesenchymal markers of EMT in the mRNA level was measured by RT-PCR. PANC-1 cells were treated with TGF-β and **1** (0.1–1 µM) for 48 h. (**B**) Quantification of each marker. Quantification was done using NIH Image J (ver. 1.51). Statistical analysis was performed using JMP 14. Data are expressed as the mean ± SE from independent experiments. * *p*< 0.05, ** *p*< 0.01.

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
