# Peer review of "Antiproliferative and Antimigration Activities of Beauvericin Isolated from Isaria sp. on Pancreatic Cancer Cells"

_molecules, 2020, doi:10.3390/molecules25194586_

Round 1
Reviewer 1 Report
I commend the work conducted by your research group on identification and isolation of active ingredients in fungi.
The work has good sound and well presented, however I have minor comment:
You need to compare the significance (at 0.05 or 0.01) of the differences among treatments in all the presented data or whenever possible to say that this compounds is more active than the other.
But in conclusion I recommend the publication after this correction.
Author Response
Thank you for the thoughtful and constructive feedback you provided regarding our manuscript.
>>The work has good sound and well presented, however I have minor comment:
You need to compare the significance (at 0.05 or 0.01) of the differences among treatments in all the presented data or whenever possible to say that these compounds is more active than the other.
But in conclusion I recommend the publication after this correction.
You raise a fair assessment. We have added the comparison of the significance at 0.05 or 0.01 in figure 3 and 4 to clarify the differences among treatments.
Reviewer 2 Report
The substantive point of interest in this paper in my reading of it is the biological activity of beauvericin, a compound known since the late 1960s. Anti proliferative and anti migratory activity are indeed relevant to potential anti cancer drugs and there is therefore a case for publication of these data. The other compounds discussed are not relevant or significant because of their total lack of activity. This is not to say that 1 - 3 are not intriguing structures that may well have some biological activity; this point would be worth following up. Since the activity of beauvericin is the key point, the paper needs a more thorough discussion of its overall biological activity, especially anticancer activity, in comparison with other compounds in the same class. I suggest that the paper requires major revision with respect to structure and content to be acceptable for publication.
On other points, it would be helpful if the authors report the NMR data in one of the standard formats (see Journal of Medicinal Chemistry, for example) to avoid ambiguity introduced by their statements. I also ask the authors to justify the use of cisplatin as a positive control; this drug requires activation in vivo and I therefore question its relevance to a simple in vitro test.
Author Response
Thank you for the thoughtful and constructive feedback you provided regarding our manuscript.
>> The substantive point of interest in this paper in my reading of it is the biological activity of beauvericin, a compound known since the late 1960s. Anti proliferative and anti migratory activity are indeed relevant to potential anti cancer drugs and there is therefore a case for publication of these data. The other compounds discussed are not relevant or significant because of their total lack of activity. This is not to say that 1 - 3 are not intriguing structures that may well have some biological activity; this point would be worth following up. Since the activity of beauvericin is the key point, the paper needs a more thorough discussion of its overall biological activity, especially anticancer activity, in comparison with other compounds in the same class. I suggest that the paper requires major revision with respect to structure and content to be acceptable for publication.
We agree with you that we need to more discussion of the activity of beauvericin, and we removed details about the elucidation of structures 1–4 and move tables 1 and 2 from the article to supporting information. Additionally, we have revised the text p. 5, lines 140–153 to reflect biological activities of beauvericin, especially anticancer activity. We did not perform comparisons between beauvericin and other compounds in the same class, so we would like to treat that as the subject of future analysis.
>> On other points, it would be helpful if the authors report the NMR data in one of the standard formats (see Journal of Medicinal Chemistry, for example) to avoid ambiguity introduced by their statements.
Thank you for providing these insights. This is a valid assessment, however, we used table format which provided on Molecules template for description of the NMR data and referenced the method of previous reports. We do hope you understand our rationale for this decision.
>> I also ask the authors to justify the use of cisplatin as a positive control; this drug requires activation in vivo and I therefore question its relevance to a simple in vitro test.
You have raised an important point. Cisplatin is surely used for measurement of in vivo cytotoxicity. However, there are many reports that cisplatin is used for cytotoxicity assay in vitro on various cancer cell line including PANC-1 cells following references shown below. Underlying mechanisms are not fully elucidated, but I assume that using cisplatin as a positive control is reasonable considering from reported evidences. We hope that you agree.
- Florea, A. M.; Busselberg, D. Cisplatin as an anti-tumor drug: Cellular mechanisms of activity, drug resistance and induced side effects. Cancers 2011, 3, 1351–1371.
- Beer, M.; Kuppalu, N.; Stefanini, M.; Becker, H.; Schulz, I.; Manoli, S.; Schuette, J.; Schmees, C.; Casazza, A.; Stelzle, M.; Arcangeli, A. A novel microfluidic 3D platform for culturing pancreatic ductal adenocarcinoma cells: comparison with in vitro cultures and in vivo xenografts. Sci. Rep. 2017, 7, DOI:10.1038/s41598-017-01256-8.
- Duan, L.; Aoyagi, M.; Tamaki, M.; Nakagawa, K.; Nagashima, G.; Nagasaka, Y.; Ohno, K.; Yamamoto, K.; Hirakawa, K. Sensitization of human malignant glioma cell lines to tumor necrosis factor-induced apoptosis by cisplatin. J. Neurooncol. 2001, 52, 23–26.
- Siddik, Z. H. Cisplatin: mode of cytotoxic action and molecular basis of resistance. Oncogene 2003, 22, 7265–7279.
Again, thank you for giving us the opportunity to strengthen our manuscript with your valuable comments and queries. We have worked hard to incorporate your feedback and hope that these revisions persuade you to accept our submission.

Round 2
Reviewer 2 Report
The authors have reacted quickly to my comments with relatively minor changes. It's not how I would have written up their results but then it's not my paper. Since the science is essentially sound, I will not stand in the way of publication.
The one new compound (2) and its relatives are largely irrelevant and I would not have included them.
Author Response
Reviewer 2
Thank you for the thoughtful and constructive feedback you provided regarding our manuscript.
>> The one new compound (2) and its relatives are largely irrelevant and I would not have included them.
We have discussed ourselves, whether the structural analysis of compounds 2–4 should be included in our article. However, compound 2 have the novelty and we consider that the elucidations of the structures of isolated compounds are important in the field of natural products chemistry. Thus, we decided to contain them in our article. We do hope you understand our rationale for this decision.
Again, we appreciate all of your insightful comments. We are hopeful that our revised focus helps to improve your opinion of work.
Sincerely,
Keiichi Matsuzaki